# Mind the Pool: Convolutional Neural Networks Can Overfit Input Size

**Bilal Alsallakh**
Voxel AI *

**David Yan**
Meta

**Narine Kokhlikyan**
Meta AI

**Vivek Miglani**
Meta AI

**Orion Reblitz-Richardson**
Meta AI

**Pamela Bhattacharya**
Meta

## Abstract

We demonstrate how convolutional neural networks can overfit the input size: The accuracy drops significantly when using certain sizes, compared with favorable ones. This issue is inherent to pooling arithmetic, with standard downsampling layers playing a major role in favoring certain input sizes and skewing the weights accordingly. We present a solution to this problem by depriving these layers from the arithmetic cues they use to overfit the input size. Through various examples, we show how our proposed spatially-balanced pooling improves the generalization of the network to arbitrary input sizes and its robustness to translational shifts.

## 1 Introduction

Convolutional neural networks (CNNs) are versatile models in machine learning. Early CNN architectures used in image classification were restricted to a fixed input size. For example, AlexNet (Krizhevsky et al., 2012) was designed to classify $224 \times 224$ images from ImageNet (Deng et al., 2009). To facilitate model comparison, this size has been adopted in subsequent ImageNet classifiers such as VGGNet (Simonyan & Zisserman, 2015) and ResNet (He et al., 2016).

The adoption of fully-convolutional architectures (Long et al., 2015; Springenberg et al., 2015) and global pooling methods (Lin et al., 2014; He et al., 2015) demonstrated how CNNs can process inputs of arbitrary size. Fully convolutional networks eliminate the use of fully-connected layers in CNN backbones, preserving 2D feature maps as the output of these backbones. Global pooling summarizes feature maps of arbitrary sizes into fixed-size vectors that can be processed using fully-connected classification layers. This ability to process inputs of varying sizes enables CNN-based classifiers to leverage their full resolution and preserving their aspect ratios.

The role of input size in CNNs has been mainly studied with respect to computational efficiency, receptive field adequacy, and model performance (Richter et al., 2021). In this paper, we study the impact of input size on the robustness and generalization of CNNs. In particular, we are interested in analyzing the sensitivity of CNNs with flexible input size to variations in this size, as illustrated in Figure 1. We demonstrate how the input size(s) used during training can strongly impact this sensitivity, and in turn, the robustness of CNNs to input shifts. We further introduce a solution to reduce this sensitivity. Our contributions are:

- Demonstrating how CNNs can overfit the boundary conditions dictated by the input size used during training [1], and identifying pooling arithmetic [2] as the culprit (Section 2).
- Introducing a modification to stride-based downsampling layers such as **maxpooling** and **strided convolution** to mitigate size overfitting (Section 3) and demonstrating how it can improve the accuracy and shift robustness of CNNs in two exemplary tasks.

In Section 4 we discuss the implications of size overfitting and link our observations with relevant findings in the literature.

---

*Work done mainly while at Meta AI.

[1] We refer to this as size overfitting for brevity.

[2] By pooling we refer to any stride-based downsampling such as maxpooling and strided convolution.

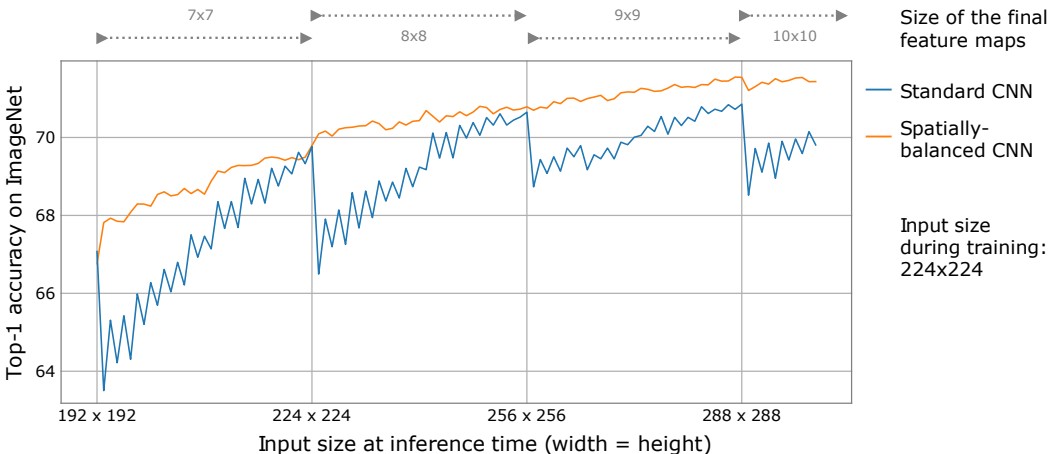

Figure 1: The ImageNet top-1 accuracy of two ResNet-18 models as a function of input size. Both models are trained on $224 \times 224$ images. The standard CNN represents the baseline available in PyTorch. Our spatially-balanced CNN mitigates periodic size overfitting.

## 2 How Do CNNs Overfit Input Size?

Consider the ResNet-18 implementation available in PyTorch (Paszke et al., 2019). We analyze the sensitivity of the provided pretrained model to variations in input size. The model is trained on ImageNet using $224 \times 224$ images. It uses global average pooling to summarize the feature maps of the last convolutional layers into scalar values.

### 2.1 Varying the input size

For the purpose of our analysis, we vary the input size from $192 \times 192$ to $299 \times 299$. This is done by simultaneously increasing the width and the height by 1 pixel, limiting the input to a square shape. This simplifies the analysis and preserves the aspect ratio used during training.

We follow the same resizing method used during training when possible. This method first resizes the image so that the smaller dimension is equal to $s = 256$. Then, the method applies a random crop of size $224 \times 224$. We use the same steps, changing mainly the crop size, and applying a centered crop instead of a random crop. This maintains the object scale to match the training images. Centered crops are typically used in the validation phase to eliminate randomness. A crop smaller than $224 \times 224$ incurs a loss of information at the periphery. A crop $s' \times s'$ larger than $256 \times 256$ would require padding. To avoid padding artifacts, we change the first step to use $s = max(s', 256)$.

The information loss in crops smaller than $224 \times 224$ and the increased object scale in crops larger than $256 \times 256$ can potentially impact the classification result of certain instances. Nevertheless, the analysis helps identify a fundamental impact of input size on CNNs, as we explain next.

### 2.2 Analyzing sensitivity to input size

For each input size, we compute the accuracy of the pretrained model on the ImageNet validation set after resizing the images as described above. Figure 1 depicts in blue the accuracy as a function of the input dimension. Both input dimensions are increased simultaneously.

The validation accuracy generally increases with the input size in the range we considered. However, there are remarkable drops in accuracy that occur periodically at an interval of 32, immediately after reaching a peak. This suggests that the model favors spec ific input sizes that correspond to these peaks, while it struggles with inputs that are 1-pixel larger in width and in height. We next demonstrate how these peaks and drops in accuracy are a byproduct of pooling arithmetic.

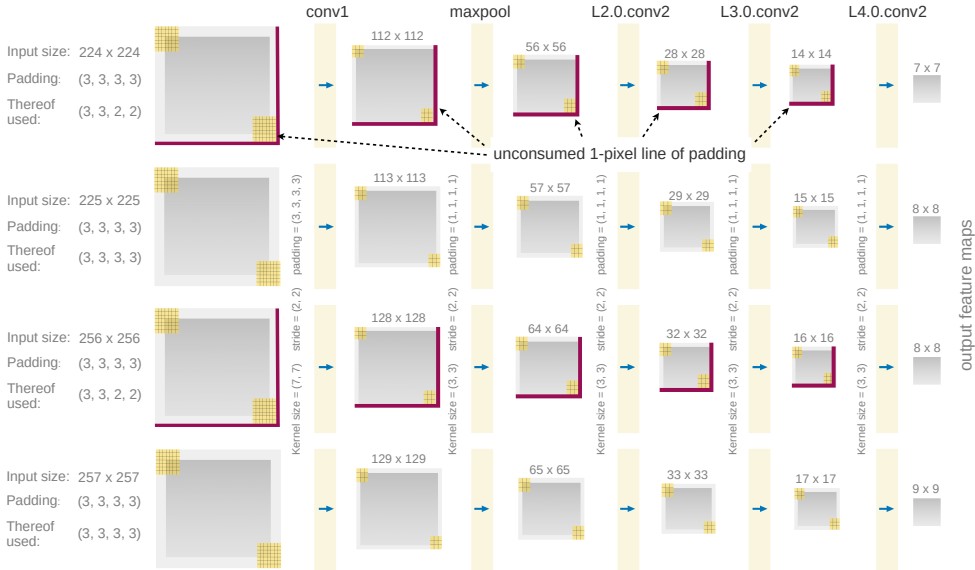

Figure 2: Illustrating how input size impacts pooling arithmetic in ResNet-18. Each column represents a stride-2 downsampling layer. An input of size $224 \times 224$ or $256 \times 256$ leads each layer to leave out 1-pixel row and column of the applied padding. In contrast, an input of size $225 \times 225$ or $257 \times 257$ causes all of these maps to have odd dimensions, leaving no padding unconsumed.

## 2.3 EXPLAINING PERIODIC SIZE OVERFITTING

To understand why the model overfits specific input sizes, we analyze how it processes inputs of different sizes. We focus on downsampling layers since the remaining layers in the convolutional backbone do not impact the size. The PyTorch implementation of ResNet-18 uses both maxpooling and strided convolution (Springenberg et al., 2015) for downsampling, with a total of five such layers. All of these layers use odd-sized kernels.

Figure 2 illustrates how four inputs of different sizes are processed by the downsampling layers. In particular, we show the output size computed by each of these layers, and whether the layer leaves parts of the padding applied to its input unconsumed. This happens when the dimensions of this input are of a different parity than the odd-sized kernel (Alsallakh et al., 2021b).

An image of size $224 \times 224$ results in an even-sized input at *each* layer, leading to unconsumed padding at the right and bottom sides of those inputs. The same happens with input images of size $256 \times 256$. In contrast, images of size $225 \times 225$ and $257 \times 257$ result in odd-sized inputs at *each* intermediate layer, leaving no padding unconsumed. Input sizes between the above edge cases result in unconsumed padding at a subset of the layers. For example, a $226 \times 226$ input impacts only the first downsampling layer while a $227 \times 227$ input impacts only the second layer. A $228 \times 228$ input impacts the first and the second downsampling layers only.

Since the model is trained on $224 \times 224$ images, it expects to consume padding only at the left and top sides of the input of each downsampling layer. The peaks of the blue plot in Figure 1 correspond to input sizes that result in the same behavior at these layers. The behavior is periodic with respect to input size. It recurs at an interval of $2^{|D|}$, where $|D|$ is the number of stride-2 downsampling layers and is equal to $|D| = 5$ in our model. The sharp drops in Figure 1 correspond to input sizes that lead to the opposite behavior: The padding is consumed at both sides of every downsampling layer, deviating significantly from the training-time behavior. These drops recur at the same interval, e.g. at $193 \times 193$, $225 \times 225$ and $257 \times 257$.

Appendix C demonstrates the above issue in various models besides ResNets such as Efficient-Net (Tan & Le, 2019), MobileNet (Sandler et al., 2018), MNASNet (Tan et al., 2019), RegNet (Radosavovic et al., 2020), ResNeXt (Xie et al., 2017) and VGGNet Simonyan & Zisserman (2015).

## 2.4 VALIDATING THE ROLE OF POOLING ARITHMETIC

To validate our observation about periodic size overfitting, we retrain the aforementioned ResNet-18 using three different input sizes: $193 \times 193$, $225 \times 225$ and $256 \times 256$. The first two sizes leave no padding unconsumed, while the third size exhibits the opposite behavior as Figure 2 illustrates.

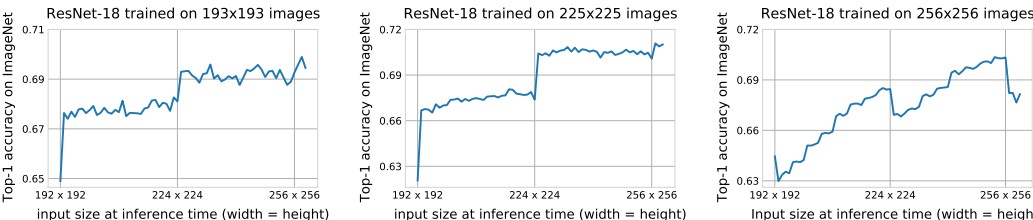

Figure 3: Sensitivity to input size of ResNet-18 trained using three different input sizes.

Figure 3 depicts the input-size sensitivity of the resulting models at inference time. It is evident that the model trained on $256 \times 256$ images exhibits the same behavior as the model trained on $224 \times 224$ ones discussed in the previous Section (Figure 1). It favors the same input sizes, and exhibits sharp drops in accuracy at the same unfavorable input sizes. Interestingly, the models trained using $193 \times 193$ and $225 \times 225$ images do not exhibit sharp drops in accuracy. Their accuracy remains relatively steady within the same interval, with significant jumps between the intervals. These jumps correspond to an increase in the size of the final feature maps, e.g. from $7 \times 7$ to $8 \times 8$ when the input size changes from $224 \times 224$ to $225 \times 225$ (Figure 2).

We aim to understand why the first two models do not exhibit periodic drops in performance, unlike the third and the original pretrained model. For this purpose, we examine the mean $3 \times 3$ kernel of each layer as suggested by Alsallakh et al. (2021a). Figure 4a illustrates how these mean kernels are computed. Figure 4b depicts the mean kernels of the original model, pretrained on $224 \times 224$ images.d Figure 4c depicts the mean kernels of the model trained on $225 \times 225$ images. It is evident that all mean kernels exhibit high symmetry about their center when trained on $225 \times 225$ images. In contrast, the mean kernels of downsampling layers or adjacent layers exhibit a strong asymmetry when trained on $224 \times 224$ images. This asymmetry is due to overexposure to zero padding at the left and top sides, caused by unconsumed padding as demonstrated by Alsallakh et al. (2021b).

Our analysis confirms the fundamental role of pooling arithmetic in the ability of CNNs to overfit input size. The input size used during training dictates the overfitting behavior. We next introduce an adjustment to CNNs that mitigates this behavior.

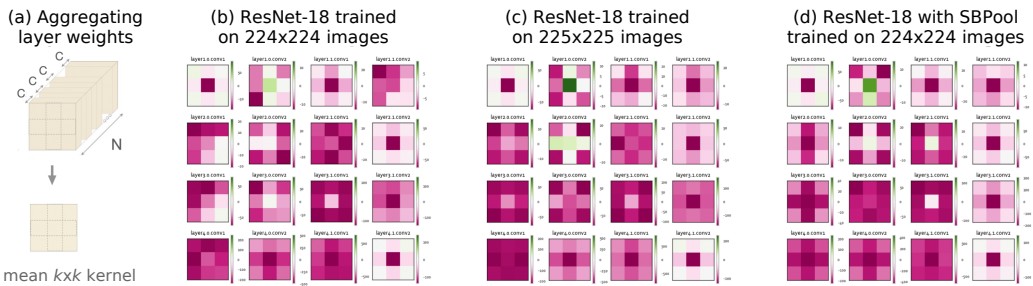

Figure 4: The impact of unconsumed padding on the learned weights in ResNet-18. (a) Computing the mean kernel in a layer as a means to examine potential spatial skewness. We show the mean $3 \times 3$ kernels of (b) the baseline model, pretrained on $224 \times 224$ images. (c) a model trained on $225 \times 225$ images, and (d) a model trained on $224 \times 224$ images with our mitigation (SBPool).

## 3  SPATIALLY-BALANCED POOLING (SBPOOL)

We propose an adjustment to downsampling layers in CNNs that aims to prevent them from overfitting the input size during training. Our adjustment is designed to deprive these layers from cues that lead to overfitting, whether the downsampling is based on pooling or on strided convolution. Our key insight is that when an input size incurs unconsumed padding at a standard downsampling layer, the latter is always located at the right and bottom sides of the layer's input (Fig 5a). In contrast, the padding at the left and top sides is always consumed. This happens over and over again as the CNN processes various training samples of the same size. This leads to overexposure to padding at the left and top sides, giving the CNN an artificial cue about that input size.

To deprive the CNN from the above-mentioned cue, we modify downsampling layers so that potentially unconsumed padding can occur at any side of the input with equal probability during training (Figure 5b). With this modification, unconsumed padding is evened out *on average* at different sides as the CNN processes various training samples. As a result, unconsumed padding does not incur spatial bias in the learned filters (Figure 4d). We call this modification SBPool.

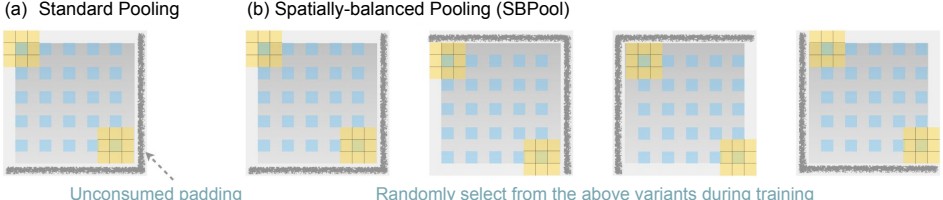

(a) Standard Pooling    (b) Spatially-balanced Pooling (SBPool)

Unconsumed padding        Randomly select from the above variants during training

Figure 5: (a) Standard downsampling: Unconsumed padding is always at the right and bottom sides of the input. (b) Our proposed downsampling: Randomly choose between four sampling grids, to warrant equal likelihood of unconsumed padding at each side.

SBPool is straightforward to implement as a wrapper of standard downsampling layers. The wrapper first computes unconsumed padding $(u_h, u_w)$ as follows: [3]

$$
\begin{aligned}
u_h &= \left(h + p_{\texttt{top}} + p_{\texttt{bottom}} - d_h \cdot (k_h - 1) - 1\right) \mod s_h \\
u_w &= \left(w + p_{\texttt{left}} + p_{\texttt{right}} - d_w \cdot (k_w - 1) - 1\right) \mod s_w
\end{aligned}
\tag{1}
$$

where $(h, w)$ are the spatial dimensions of the layer's input, $(p_{\texttt{top}}, p_{\texttt{left}}, p_{\texttt{bottom}}, p_{\texttt{right}})$ is the padding intended to be applied to that input, $(k_h, k_w)$ is the kernel size, and $(d_h, d_w)$ is the kernel dilation, and $(s_h, s_w)$ are the stride factors. The padding applied is then modified to randomize at which side the unconsumed padding $(u_h, u_w)$ is effectively incurred:

$$
\begin{aligned}
p_{\texttt{top}} &\leftarrow p_{\texttt{top}} - \mathrm{randInt}\left([0, u_h]\right) \\
p_{\texttt{left}} &\leftarrow p_{\texttt{left}} - \mathrm{randInt}\left([0, u_w]\right)
\end{aligned}
\tag{2}
$$

where $\mathrm{randInt}()$ is a function that returns an integer within the given range uniformly at random.

The above modification to the padding applied allows the convolutional kernel to randomly assume any of the starting positions and corresponding sampling grids illustrated in Figure 5b. Appendix A provides an algorithmic outline of the described modification. Note that when parallel branches perform downsampling as in skip connections, we ensure the sampling grids they assume are aligned (refer to Appendix E for illustration).

At inference time, we assign each downsampling layer a fixed sampling grid to ensure deterministic evaluation results. We select different grids for these layers to minimize the distortion of the receptive field as we discuss in Section 4.

We demonstrate how SBPool mitigates size overfitting in two exemplary tasks that involve different CNN architectures and datasets. Appendix D provides additional examples on other architectures.

---

[3]The equation is adapted from the documentation of MaxPool2d in PyTorch

## 3.1 IMAGE CLASSIFICATION

We retrain the ImageNet classification model described in Section 2 with SBPool using the same hyperparameters. Both the baseline model and the new model are trained on $224 \times 224$ images, and their corresponding feature maps are hence of equal sizes. Accordingly, both models use the same compute power (FLOPS) during training. [4]

Figure 1 depicts the top-1 accuracy of both models as a function of the input dimension. For each input size, we resize the images in the validation set as described in Section 2.1. Table 1 lists the accuracy for selected sizes. It is evident that the spatially-balanced model does not suffer from sharp drops in accuracy, unlike the baseline model. With sizes favorable for the baseline model, marked with (*), both models have comparable performance. However, at unfavorable sizes the spatially-balanced model significantly outperforms the baseline model.

Table 1: Top-1 accuracy of ResNet-18 on ImageNet for different input sizes at the inference phase.

| Input size | $224 \times 224^*$ | $225 \times 255$ | $256 \times 256^*$ | $257 \times 257$ | $288 \times 288^*$ | $289 \times 289$ |
|---|---|---|---|---|---|---|
| Baseline | **69.76** | 66.53 | 70.63 | 68.74 | 70.83 | 68.53 |
| CNN with SBPool | 69.64 | **70.10** | **70.81** | **70.73** | **71.54** | **71.21** |

Figure 4d depicts the mean kernels of the spatially-balanced model. It is evident that SBPool mitigates the asymmetry observed in the baseline model (Figure 4b). This improves the shift robustness of the model, as we demonstrate in Appendix C. The results generalize to other model families such as MobileNet and VGGNet as we demonstrate in Appendix D.

## 3.2 SEMANTIC SEGMENTATION

Dilated residual networks (Yu et al., 2017) serve as a simple architecture for semantic image segmentation. We select a model pretrained by the authors [5] on the Cityscapes dataset (Cordts et al., 2016). The model contains three downsampling layers ($|D| = 3$). Its weights are initialized using an equivalent image classification model trained on ImageNet. Subsequently, the model is trained on $896 \times 896$ image patches randomly sampled from the Cityscapes training set. In the testing phase, the model is applied to the entire image, whose size is $2048 \times 1024$.

To obtain a spatially-balanced segmentation model, we first retrain the classification model used for initialization under SBPool, and then retrain the segmentation model. Figure 6 depicts the performance of both the baseline model and the spatially-balanced model as functions of input size. For each size, we crop the test set images along with the corresponding ground truth images into centered squares of that size and compute the mean Average Precision (mAP) of both models on those images.

It is evident in Figure 6 that the baseline model exhibits periodic size overfitting at an interval of $2^{|D|} = 8$. This behavior is mitigated in the spatially-balanced model whose performance increases steadily with input size, exhibiting minor fluctuation. Moreover, the latter model outperforms the baseline by a visible margin. Table 2 lists the mAP of both models for different input sizes, including the ones used originally in the training phase ($896 \times 896$) and in the validation phase ($2048 \times 1024$). The performance of the baseline model drops significantly with the unfavorable sizes of $201 \times 201$ and $897 \times 897$.

Table 2: Performance (mAP) of segmentation models on CityScapes with different crop sizes.

| Input size (centered crop) | $200 \times 200$ | $201 \times 201$ | $896 \times 896$ | $897 \times 897$ | $2048 \times 1024$ |
|---|---|---|---|---|---|
| Baseline | 41.180 | 39.380 | 66.63 | 66.19 | 68.00 |
| CNN with SBPool | **41.65** | **42.28** | **67.65** | **67.72** | **68.37** |

---

[4]This does not hold when training on $225 \times 225$ images as done by Alsallakh et al. (2021b) since the internal feature maps will be larger than the ones with $224 \times 224$ input images as illustrated in Figure 2

[5]The model name is DRN-D-22 and is available at `http://go.yf.io/drn-cityscapes-models`

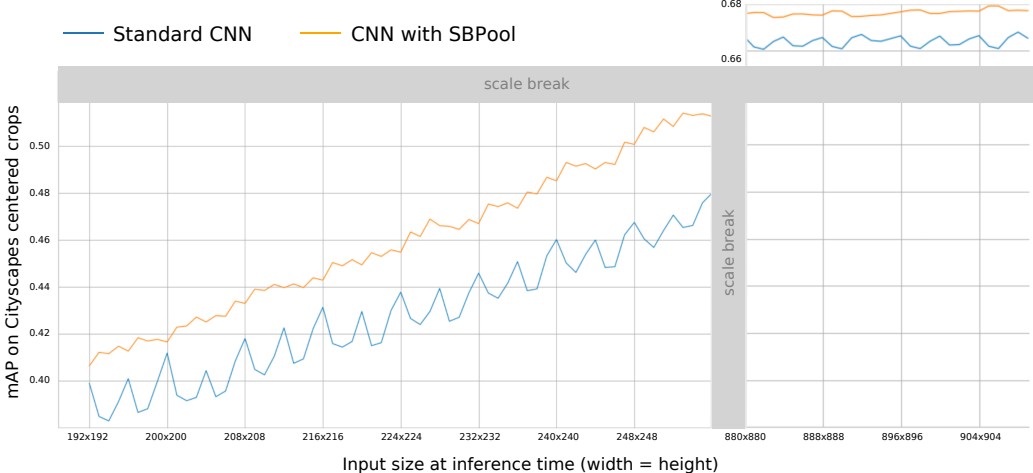

Figure 6: The mean Average Precision (mAP) of two Cityscapes semantic segmentation models of the same architecture (Yu et al., 2017), as a function of input size. The baseline model is pretrained by the original authors [5]. The spatially-balanced model is retrained under SBPool.

## 4 DISCUSSION

Size overfitting has been observed in certain language models trained on a fixed input size (Wang & Chen, 2020). We have demonstrated that, unless mitigated, CNNs are prone to overfitting input size, which can skew the learned weights.

**Why Does Size Overfitting Matter?** Even if the input size is identical at training and inference time, it is worth to mitigate size overfitting. This mitigation improves shift invariance as we show in Appendix C. Oftentimes, the learned weights are used to initialize other models, trained on other tasks and datasets. For example, weights pretrained on ImageNet with $224 \times 224$ images are widely used for this purpose, even when the downstream task uses a different or a variable input size. This makes the pretrained weights a sub-optimal starting point, as our results demonstrate (Figure 6).

**Sources of Inspiration** SBPool is inspired by practical solutions to related problems. Notably, to prevent flat-head syndrome in infants, the head orientation should be varied when in supine position (Xia et al., 2008). Likewise, spaceships need continuous rolling to avoid overheating of the side exposed to the Sun, and automobile tires need to be rotated on a regular basis. Wu et al. (2019) employ a similar idea to avoid information erosion when using $2 \times 2$ convolutional kernels. Their solution divides the feature maps of each layer into four groups that correspond to the variations in Figure 5. The key difference in our work is that SBPool injects variation across *different samples* not within the feature maps computed for one sample. Accordingly, it serves a different purpose and is only needed for downsampling layers during training.

### 4.1 IS PADDING THE CULPRIT?

Padding is known to impact CNN's spatial invariance (Kayhan & van Gemert, 2020; Islam et al., 2020). We examine whether it is the only cue CNNs can use to overfit input size.

**What If Other Padding Methods Were Applied?** The downsampling layers in the examples presented so far use zero padding. Other padding methods were shown to alleviate feature map artifacts incurred by salient lines of zero (Alsallakh et al., 2021b). We analyzed size overfitting in various ResNet models trained on ImageNet under different padding methods, including Partial Convolution (Liu et al., 2018). All of these models are trained on $224 \times 224$ images, which incur unconsumed padding at every downsampling layer. Figure 7 depicts two examples, showing how the models overfit the input size. This demonstrates how one-sided unconsumed padding can provide CNNs with sufficient cues about the input size, regardless of the padding method.

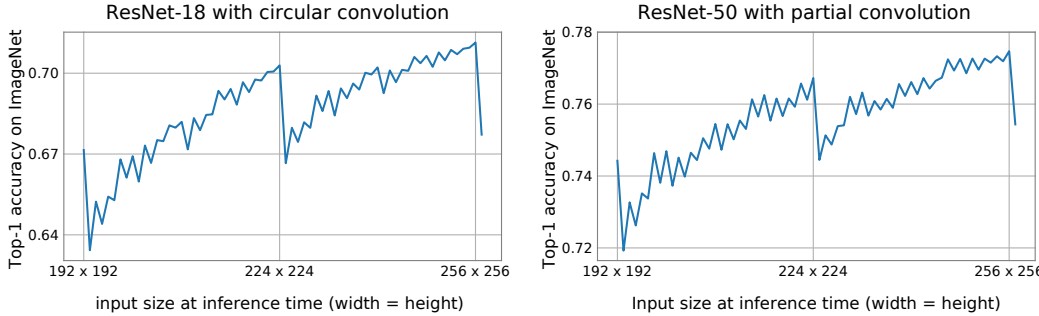

Figure 7: Sensitivity to input size of two ResNet models trained under different padding methods: circular and PartialConv (Liu et al., 2018). Both models are trained on $224 \times 224$ images.

**What if No Padding Were Applied?** Downsampling layers often apply padding to avoid marginalization of the periphery, especially when their kernels are $3 \times 3$ or larger. When no padding is applied, Eq 1 computes unconsumed parts of the layer's input that represent information erosion. Appendix D provides an example with VGGNet which applies no padding during downsampling. Nevertheless, VGGNet overfits the input size. Furthermore SBPool mitigates this overfitting by randomly selecting at which side the information erosion occurs. This suggests that information erosion provides CNNs with cues to overfit the input size. To further investigate such cues, we next analyze how different input sizes impact the receptive field.

## 4.2 IMPACT ON RECEPTIVE FIELD

Richter et al. (2021) demonstrate how a mismatch between input size and the receptive field impacts object recognition in CNNs, focusing on object scale. Jang et al. (2022) proposed a data-driven approach to optimize the receptive field. Here we focus on how *pooling arithmetic* impacts the receptive field, making it sensitive to small changes in input size.

Figure 8 depicts the receptive field of a ResNet-50 model under various conditions, following the visualization method proposed by Alsallakh et al. (2021c). A $225 \times 225$ input incurs no unconsumed padding, leading to a well-centered receptive field in the input space. A $224 \times 224$ input incurs unconsumed padding at every downsampling layer. This in turn shifts the center of the receptive field in the baseline model to the top left corner. By applying SBPool, unconsumed padding is balanced at opposing sides, recovering the alignment between the receptive field and the input. The misalignment in the middle plot of Figure 8 provides a cue which enables CNNs to periodically overfit sizes that exhibit the same misalignment. Shocher et al. (2020) observed a similar misalignment between the input and the output in CNNs and proposed a special type of convolutional layers as a mitigation.

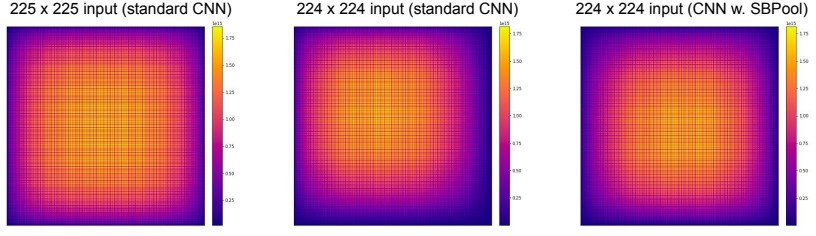

Figure 8: The receptive field of ResNet-50 under various conditions. Without SBPool, a $224 \times 224$ input causes misalignment of the field's center in the input space, as the middle plot demonstrates.

## 4.3 POOLING TECHNIQUES IN CNNS

Subsampling, pooling, and strided convolution techniques are ubiquitous in modern CNNs, and have received a significant amount of research attention. Boureau et al. (2010) explored mixing max and

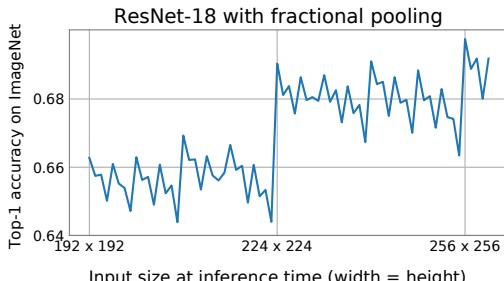 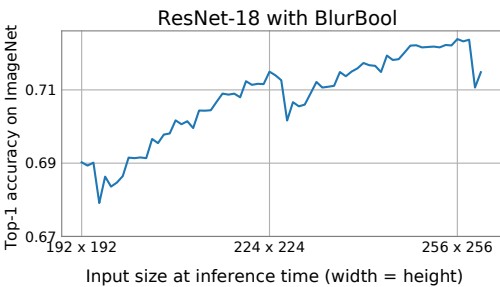

Figure 9: Sensitivity to input size under different pooling methods, Fractional Pooling (Graham, 2014) and BlurPool (Zhang, 2019). Both models are trained on $224 \times 224$ images.

average pooling, with extensions proposed to include learned weights (Lee et al., 2016), stochasticity (Yu et al., 2014), or both (Kobayashi, 2019b). These extensions aim to improve on the deficiencies of max and average pooling, namely detail loss, introduction of artifacts, and overfitting. A family of related methods involve randomly sampling from pooling regions, and tend to confer a regularizing effect and to increase shift invariance (Zeiler & Fergus, 2013; Zhai et al., 2017; Kobayashi, 2019a). At inference time, such stochastic pooling methods typically use some form of averaging over the probabilities used during training.

Another area of work involves subsampling operators which permit arbitrary input-to-output ratios (Graham, 2014; Jang et al., 2022; Shocher et al., 2020). Spectral pooling methods (Rippel et al., 2015; Zhang & Ma, 2020; Zhang, 2021) cast subsampling as a cropping in the frequency domain, allowing any output size while retaining maximum information. Recently, Riad et al. (2022) extended spectral pooling to allow learnable strides. Decomposing images into frequency components can also be seen in wavelet pooling (Williams & Li, 2018) and in LiftPool (Zhao & Snoek, 2021).

A number of pooling techniques have been proposed to improve translation invariance. BlurPool (Zhang, 2019) reduces the aliasing incurred by strided operations by applying a Gaussian blur. Hossain et al. (2021) uses a similar approach, additionally allowing the standard deviation to be learnable to control the strength of blurring. Xu et al. (2021) and Chaman & Dokmanic (2021) enforce translation equivariance using sampling techniques tailored for that purpose.

Besides max-pooling, average pooling, and strided convolution, we analyze input-size overfitting under two popular pooling techniques: Fractional Pooling (Graham, 2014) and BlurPool. For this purpose, we employ these techniques in two ImageNet classification models, trained on $224 \times 224$ images. The first model is based on VGGNet, where we replace max-pooling layers with fractional pooling layers having an input-output ratio of 0.5. The second model is a ResNet-18 pretrained by Zhang (2019). Figure 9 depicts the performance of each model as a function of input size. It is evident that the both models exhibit periodic size overfitting. Interestingly, the BlurPool model exhibits a delay in the performance drops, compared with the baseline model in Figure 1. As a future work, we are interested in understanding potential overfitting behavior of various pooling methods and how applicable SBPool could be with these methods.

## SUMMARY

We demonstrated how pooling arithmetic makes CNNs susceptible to overfitting the input size. The overfitting behavior is periodic, favoring sizes that induce similar boundary conditions at pooling layers to the ones encountered during training. We demonstrated how overfitting can skew the learned weights and impact shift invariance in CNNs. We presented a technique to mitigate overfitting by preventing pooling layers from developing specific arithmetic patterns during training. Our technique can be incorporated into standard pooling and downsampling layers. Through various experiments, we demonstrate how this technique improves the robustness of CNNs to translational shifts and to changes in the input size. This helps make the learned representation more generic for downstream tasks and can confer significant improvement in their accuracy.

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

## A  SBPool Algorithm

The following listing demonstrates how SBPool wraps existing downsampling layers to spatially-balance the downsampling results. Refer to Eq 1 and Eq 2 for explanation of the layer parameters involved (denoted by $L.variable$) and of the variables introduced ($u_h$ and $u_w$).

---

**Algorithm 1:** Spatially-Balanced Pooling

**Input:** $L$: The downsampling layer, e.g. maxpooling or strided convolution
**Input:** $X$: The layer's input: a tensor of shape (#samples, #channels, height, width)
**Output:** Y: Downsampling result

$u_h \leftarrow \big(X.height + L.p_{\text{top}} + L.p_{\text{bottom}} - L.d_h \cdot (L.k_h - 1) - 1\big) \mod L.s_h$

$u_w \leftarrow \big(X.width + L.p_{\text{left}} + L.p_{\text{right}} - L.d_w \cdot (L.k_w - 1) - 1\big) \mod L.s_w$

$L.p_{\text{top}} \leftarrow L.p_{\text{top}} - \text{randInt}\big([0, u_h]\big)$

$L.p_{\text{left}} \leftarrow L.p_{\text{left}} - \text{randInt}\big([0, u_w]\big)$

**if** $L.p_{top} < 0$ **then**
$\quad X \leftarrow X[:,:,-L.p_{\text{top}}:,:]\,;\ L.p_{\text{top}} \leftarrow 0$ /* balance vertical erosion     */

**if** $L.p_{left} < 0$ **then**
$\quad X \leftarrow X[:,:,:,-L.p_{\text{left}}:]\,;\ L.p_{\text{left}} \leftarrow 0$ /* balance horizontal erosion */

$Y \leftarrow L(X)$

Reset $L.p_{\text{top}}$ and $L.p_{\text{left}}$ to their original values

---

## B    SIZE OVERFITTING EXAMPLES

Here we demonstrate size overfitting in various models trained on different datasets and tasks.

### B.1    SCENE CLASSIFICATION

We examine two ResNet models trained on Places365 (Zhou et al., 2017) and provided by the dataset curators at `https://github.com/CSAILVision/places365`. The images were resized to $224 \times 224$ during training. The models have five downsampling layers that use odd-sized kernels. Figure 10 demonstrates how both models exhibit periodic size overfitting at an interval of 32, similar to the ImageNet example presented in Section 2.

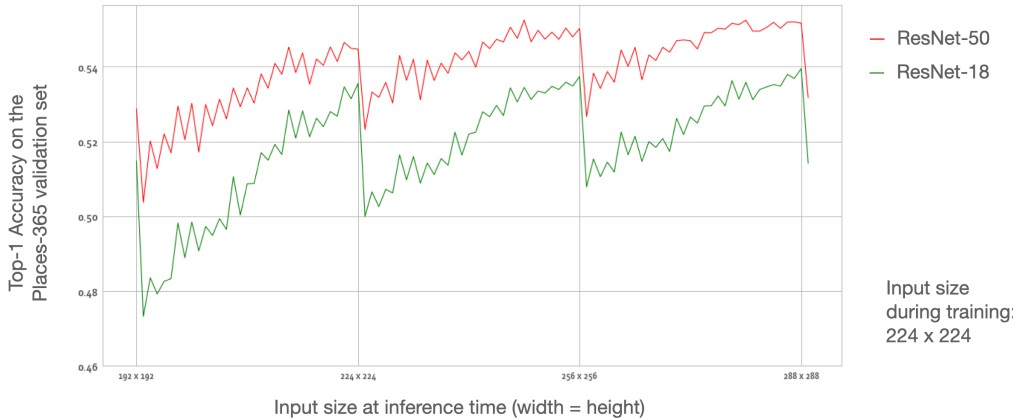

Figure 10: The accuracy of two scene classification models, trained on Places-65, as a function of input size. Both models are trained on $224 \times 224$ images and contain $|D| = 5$ downsampling layers. Both models exhibit periodic size overfitting at an interval of $2^{|D|} = 32$.

### B.2    CITYSCAPES SEMANTIC SEGMENTATION

We select two additional Cityscapes models pretrained by the author of dilated residual networks: DRN-D-38 and DRN-D-105, available at `http://go.yf.io/drn-cityscapes-models`. Both models contain three downsampling layers. Figure 11 demonstrates how both models exhibit periodic size overfitting at an interval of $2^3 = 8$, similar to the example presented in Section 3.2.

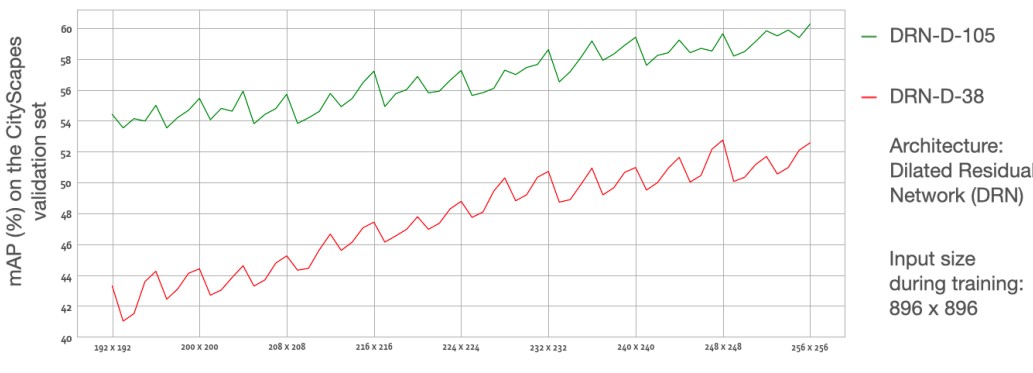

Figure 11: The mAP of two semantic segmentation models on CityScapes, as a function of input size. Both models are pretrained by Yu et al. (2017) on $896 \times 896$ patches and contain $|D| = 3$ downsampling layers. Both models exhibit periodic size overfitting at an interval of $2^{|D|} = 8$.

## B.3 IMAGENET CLASSIFICATION

We examine five ImageNet classification models available in PyTorch (Paszke et al., 2019) and pretrained on $224 \times 224$ images. Each model contains five downsampling layers ($|D| = 5$) that use odd-sized kernels. Figures 12 and 13 demonstrates how these models exhibit periodic size overfitting at an interval of 32. It is noticeable that with MNASNet (Tan et al., 2019), there are intermediate peaks and drops, e.g., when the input size is $200 \times 200$, $208 \times 208$, or $216 \times 216$. These peaks correspond to partial agreement between the unconsumed padding patterns between these sizes and the size used during training (see Figure 2).

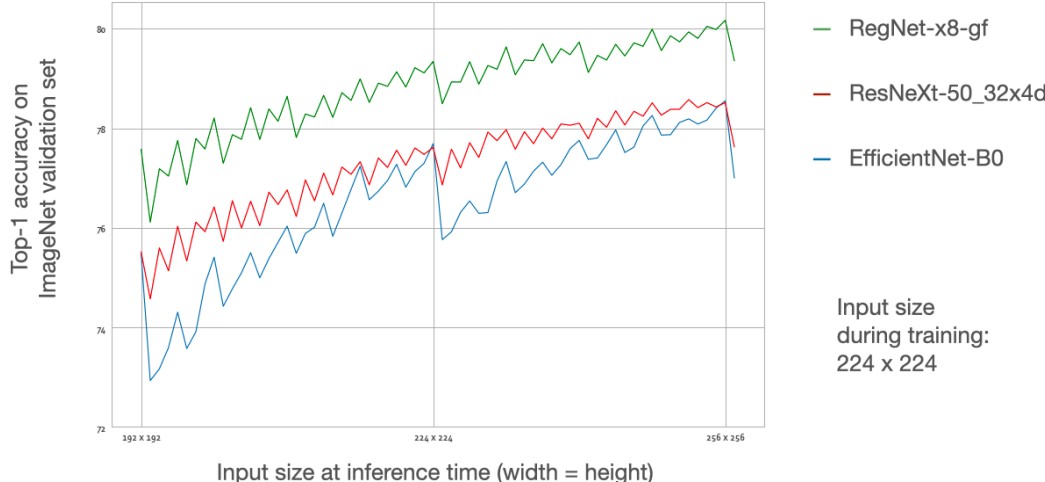

Figure 12: The top-1 accuracy of three ImageNet classifiers as a function of input size: Efficient-Net (Tan & Le, 2019), RegNet (Radosavovic et al., 2020), and ResNeXt (Xie et al., 2017). Each model contains five downsampling layers and exhibits periodic size overfitting at an interval of 32.

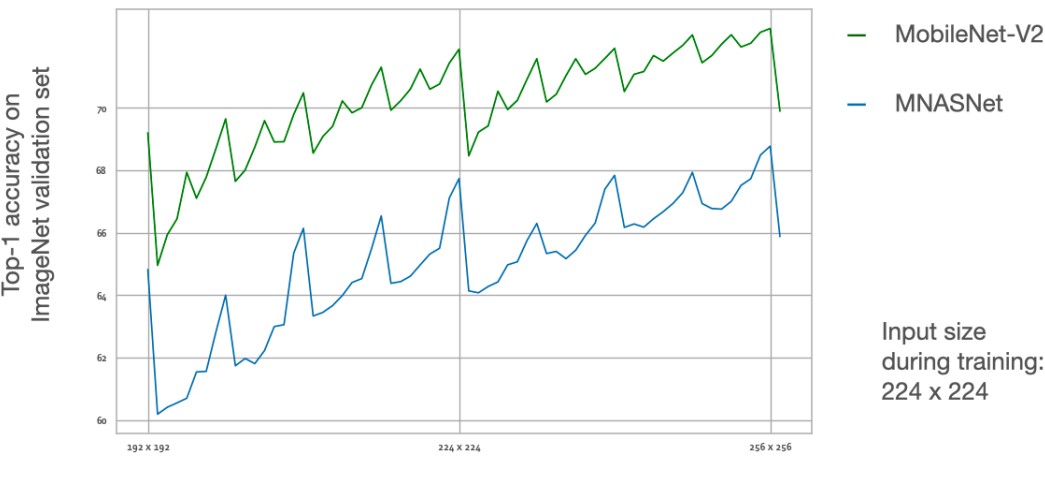

Figure 13: The top-1 accuracy of two ImageNet classification models, MobileNet (Sandler et al., 2018) and MNASNet (Tan et al., 2019), as a function of input size. The models contain $|D| = 5$ downsampling layer and hence exhibit periodic size overfitting at an interval of $2^{|D|} = 32$. Intermediate peaks are also visible within each interval. These peaks correspond to partial agreement in unconsumed padding patterns between the respective input size and the one used during training.

## C  SIZE OVERFITTING AND SHIFT CONSISTENCY

We compute the shift consistency metric proposed by (Zhang, 2019) for the models discussed in Section 3 with a range of input sizes. Figure 14 shows the consistency both of the standard ResNet-18 and of the model trained on ImageNet under SBPool as functions of input size. It is evident that SBPool improves the consistency for input sizes that do not match the preferred ones.

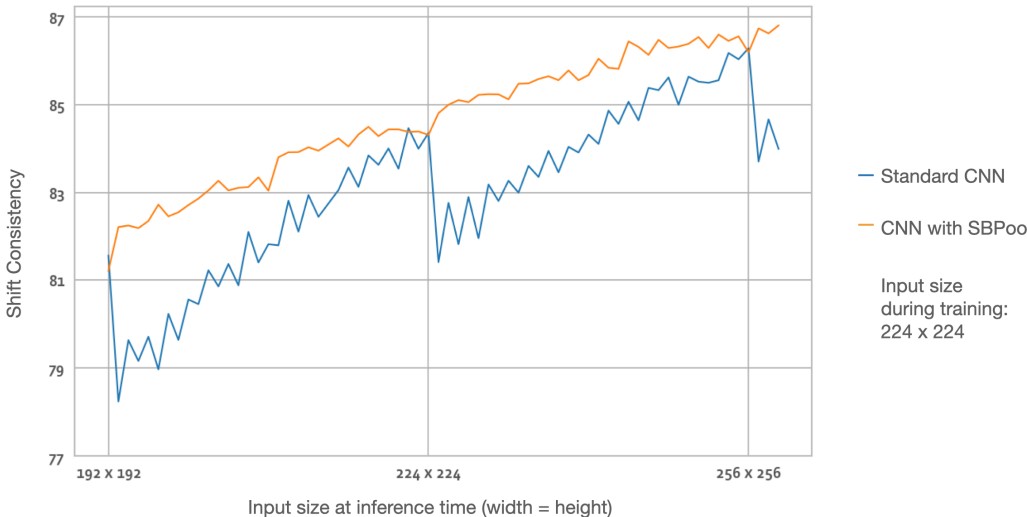

Figure 14: The shift consistency of ResNet-18 on ImageNet classification as a function of input size, with and without SBPool.

To compute the consistency in case of semantic segmentation, we generalize the metric proposed by (Zhang, 2019) beyond image classification. For this purpose, we use the segmentation result of the full input size at inference time as a reference, and compute how consistent these results are with the ones computed for random crops of a specific size. Figure 14 shows the consistency of the standard DRN model on CityScapes, compared with the model trained under SBPool as functions of the crop size. It is evident that SBPool improves the consistency for various input sizes by a significant margin.

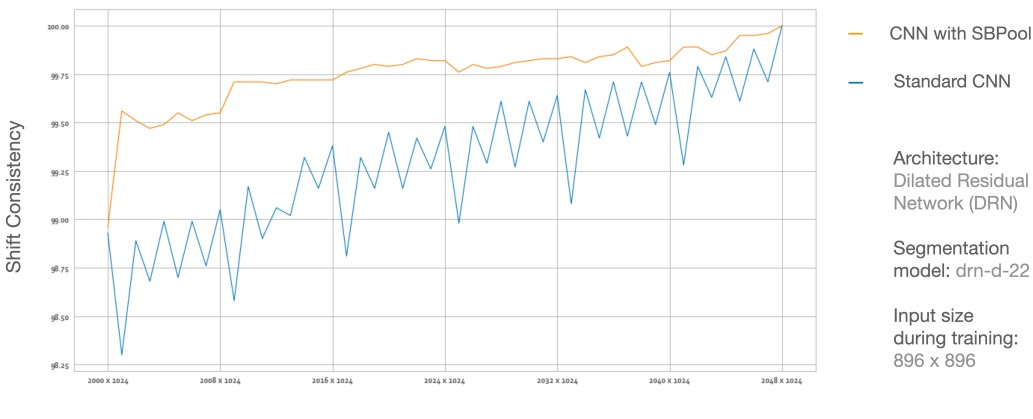

Figure 15: The shift consistency of the DRN-D-22 segmentation model computed on CityScapes with and without SBPool, as a function of input size.

## D SBPOOL EXAMPLES

We demonstrate SBPool on two additional models, besides the ones in Section 3.

### D.1 MOBILENET-V2

We retrained the MobileNet-V2 (Sandler et al., 2018) model available in PyTorch for ImageNet classification under SBPool, using the same training recipe. Figure 16 plots the size sensitivity of both the retrained model and the one provided by PyTorch. As expected, the retrained model improves the generalization to varying input size.

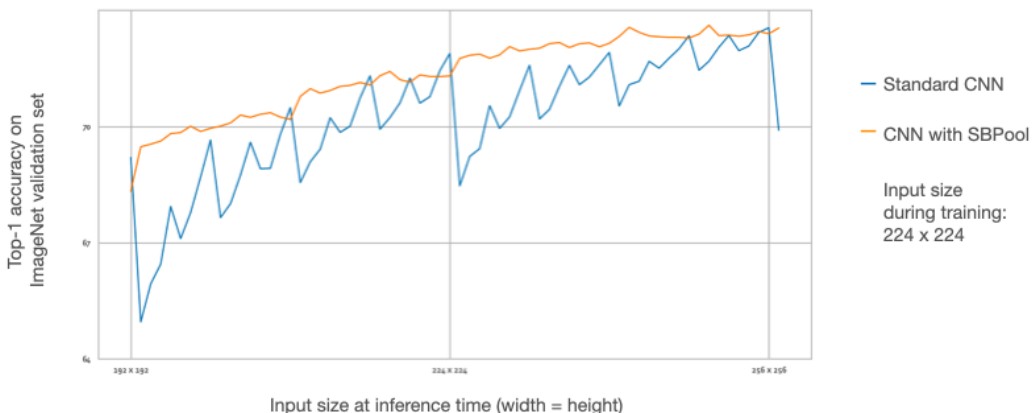

Figure 16: The top-1 accuracy of MobileNet-V2 on ImageNet classification as a function of input size, with and without SBPool.

### D.2 RESNET-50

We retrained the ResNet-50 (He et al., 2016) model available in PyTorch for ImageNet classification under SBPool, using the same training recipe. Figure 16 plots the size sensitivity of both the retrained model and the one provided by PyTorch. As expected, the curve corresponding to the retrained model exhibits no sharp drops in performance and less oscillation, unlike the pretrained model. Moreover, the mean $3 \times 3$ kernels of the retrained model show high symmetry, while certain mean kernels in the pretrained model are highly asymmetric.

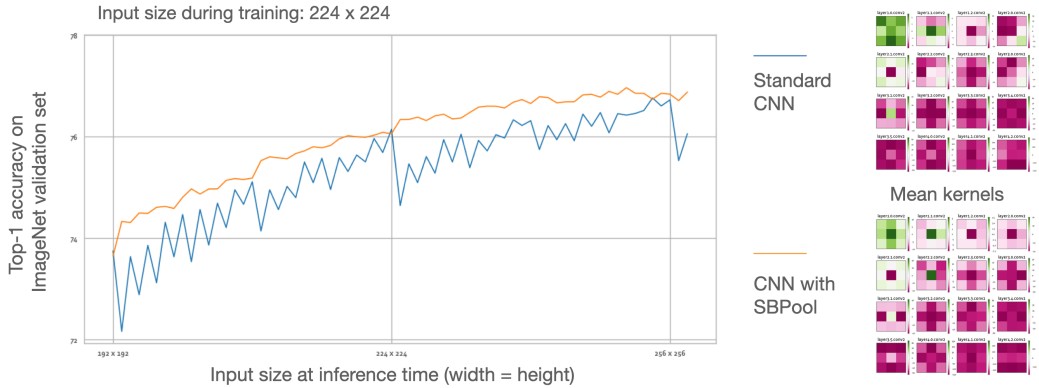

Figure 17: The top-1 accuracy of ResNet-50 on ImageNet classification as a function of input size, with and without SBPool.

### D.3    VGGNET

VGGNet models for ImageNet classification contain five maxpooling layers. These layers use $2 \times 2$ kernels and apply no padding, unlike ResNet models. Accordingly, an input of size $224 \times 224$ is evenly processed at each of these layers. Moreover, An input of size $223 \times 223$ leads to uneven erosion at each pooling layer: 1-pixel lines at the bottom and right sides are ignored by the layer.

We trained two VGG-11 models on ImageNet with $223 \times 223$ as input size, once with SBPool and once without SBPool. Figure 18-left shows the sensitivity of both models to input size at inference time. As expected, without SBPool the CNN overfits the input size: It shows significant drops when the size is $224 \times 224$ or $256 \times 256$, as both sizes prevent erosion, unlike the size for training. In contrast, with SBPool the CNN does not overfit the input size, instead showing a steady improvement in the top-1 accuracy as the input size increases, with minimal oscillation.

Figure 18-right compares the sensitivity of the SBPool model with the pretrained model available in PyTorch, which is trained on $224 \times 224$ images. While the latter model does not show sharp drops, it does prefer the size used during training, showing a significant jump in accuracy at this size as well as at $256 \times 256$. Like the training size, the latter size prevents erosion at pooling layers.

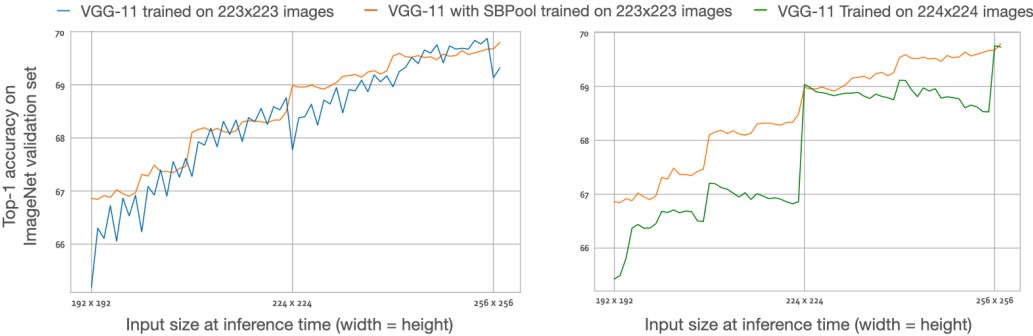

Figure 18: The top-1 accuracy of three VGG-11 models on ImageNet classification as a function of input size. The left plot compares two of these models trained on $223 \times 223$ images with and without SBPool. The right plot compares the SBPool model with the baseline model available in PyTorch.

## E    SBPOOL WITH PARALLEL BRANCHES

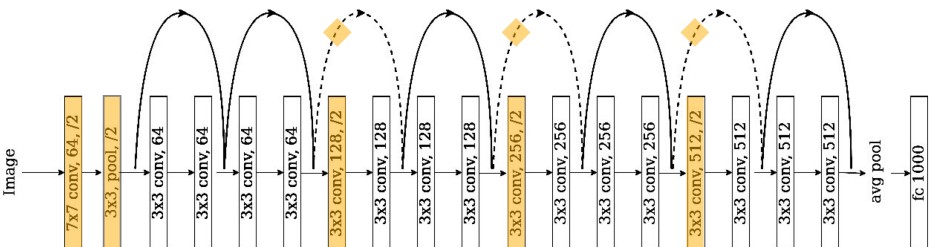

Figure 19: A diagram of ResNet-18 showing the main branch as well as skip connections. Downsampling layers are highlighted in orange. The diamonds represent downsampling operations that take place in skip connections, parallel to their counterparts in the main branch. We ensure that these operations use the same sampling grids SBPool assigns to their counterparts in the main branch. (Diagram adapted from He et al. (2016)).

