# OpenReview forum: "Mind the Pool: Convolutional Neural Networks Can Overfit Input Size"
_ICLR.cc/2023/Conference — ICLR 2023 poster_

### Official Review · Reviewer_b1SZ · 2022-10-24

**Confidence:** 4
**Correctness:** 4
**Technical Novelty And Significance:** 2
**Empirical Novelty And Significance:** 3
**Recommendation:** 8

**Clarity, Quality, Novelty And Reproducibility:**

The paper is well-written and clear. The quality of the work is good.

The novelty of this work is limited (both the problem and solution are quite simple), but this is compensated by an exhaustive set of experiments and thorough discussion of the problem, which still provides value to the community.

As the experiments mostly use off-the-shelf models which can be readily retrained on different input sizes, I imagine the results to be easily reproducible.

**Strength And Weaknesses:**

I believe this paper highlights a simple problem, explains the cause, and proposes a simple solution. Although the problem and solution are both simple, the authors did a good job at being very exhaustive in their experiments and discussion of related work, which I believe gives the paper sufficient content to be considered for acceptance.

Most of my concerns are relatively minor. One minor issue I see is that the paper repeatedly talks about "unused padding" or "pooling" as the reasons for overfitting. But the input size overfitting is unrelated to padding, which the authors acknowledge in section 4.1. The term "pooling" on the other hand seems a bit confusing, because that is a word I associated with, e.g., average- or max-pooling, and not with a regular downsampling layer (i.e., a kernel which uses a stride greater than 1). Perhaps the authors could clarify these nuances earlier on.

Minor comments:

* There's a bracket missing in section 2.4 after "(Figure 1"
* In section 2.2 it says that "it is not immediately obvious why the other peaks recur at an interval of 32". I read this statement as "we don't know why this happens", but then in section 2.3 it turns out that the authors do know. Perhaps this sentence can simply be deleted.
* The term "overfitting" is a bit confusing, because the model's best results are often not for input sizes that it was trained on (e.g., in figure 1 the best results are for 256 or 288, not for 244), so it's strange to say that it has overfit to the training input size. I guess the precise statement would be that the model has overfit to the border conditions caused by downsampling? Perhaps the wording should reflect this (e.g., in the "Our contributions" part in section 1).

**Summary Of The Paper:**

This paper looks at how convolutional neural networks can overfit on the input size. This is first shown experimentally (figure 1) and then the authors hypothesize that it is due to downsampling (figure 2). Simply put, there is no way to apply a kernel of size 3 (for example) with a stride of 2 over an input of size 4 in such a way that it is not biased. They validate this hypothesis (figures 3 and 4) experimentally.

In section 3 the authors propose a solution: In cases where the kernel size is not a divisor of the padded input size, they divide the padding between both sides randomly. They show experimentally that this solution reduces sensitivity to input sizes and finish the paper with a detailed discussion of related work.

**Summary Of The Review:**

A simple solution to the simple problem of downsampling layers in neural networks overfitting to the boundary conditions, but given that this is a well-written paper with an exhaustive set of experiments, I think this paper should be considered for acceptance.

---

> ### Author Response · Authors · 2022-11-16
> **Clarifying the terminology upfront; the role of unused padding**
>
> We thank our reviewer for the good summary and for the helpful remarks on our submission. We agree that our usage of the terms pooling and size overfitting might be confusing and have updated the manuscript to clarify what we mean upfront in the list of contributions:
> - “Demonstrating how CNNs can overfit the boundary conditions dictated by the input size used during training” (Footnote: we refer to this as size overfitting).
> - “[..] and identifying pooling arithmetic as the culprit”  (Footnote: By pooling we refer to any stride-based downsampling such as maxpooling and strided convolution).
> - We highlighted maxpooling and strided convolution as two examples of downsampling.
>
> Regarding the repeated use of "unused padding", we agree that this condition is specific to pooling / stride-based downsampling layers, and that the general use of padding in itself is not the culprit. The culprit is rather the boundary condition dictated by all elements of pooling arithmetic as Reviewer JXFi noted. It can manifest in several cues besides unused padding such as the misalignment of the receptive field or one-sided feature-map erosion in padding-free pooling layers (as in VGG models with 223x223 inputs). We can elaborate on this in the discussion Section if recommended by our reviewer.
>
> We appreciate our reviewer’s further comments on a few presentation issues, we have updated the manuscript accordingly. We hope we could sufficiently address the issues raised by our reviewer and appreciate guidance in case further changes can help improve our manuscript.

---

### Official Review · Reviewer_L9hd · 2022-10-27

**Confidence:** 5
**Clarity, Quality, Novelty And Reproducibility:** Highly reproducible.
**Correctness:** 4
**Technical Novelty And Significance:** 4
**Empirical Novelty And Significance:** 4
**Recommendation:** 6

**Strength And Weaknesses:**

My summary of the paper notes the strengths of paper. It is an extremely simple idea that works very well in practice. Experiments and discussion are through along with the exposition which helped set up the problem in the first place.

While I do not have technical qualms, I am extremely disappointed by the inaccessible plots and figures. They are of extremely poor quality and would benefit from a revision with every single plot and figure being redone and rendered with vector graphics.

I strongly urge the authors to fix this and I would be willing to give it a higher score after that and after discussions with other reviewers.

**Summary Of The Paper:**

This paper investigates a problem I would not have thought about -- CNNs overfitting to input size due to the pooling arithmetic involved -- this in itself is an extremely exotic thought! The authors show that due to the unconsumed padding, often the networks trained on specific input sizes will not generalize or extrapolate They proceed to propose a simple solution called Spatially-Balanced Pooling (SBPool) that mitigates this phenomenon, by randomly alternating the location of the unconsumed line of padding. The results are extremely strong for ImageNet classification and Semantic segmentation across architectures. They conclude with a nice analysis and discussion alongside strong baselines which fail to fix the phenomenon.


(The brevity of the review should not be taken negatively, it is due to the clarity of the paper in its ideas, hypothesis and experimentation).



**Summary Of The Review:**

See above. I really want the authors to submit a revision with fixed figures. It reflects poorly to have this as a major issue when the idea and the paper itself are extremely well done!

---

> ### Author Response · Authors · 2022-11-18
> **Making the figures accessible**
>
> We thank our reviewer for the encouraging comments and for the good suggestion to make the figures accessible.
>
> We have revised most figures in the main draft to use vector graphics, and are working on redoing Figure 2 which we commit to improve as well.
> With that we hope the revised figures now are of a decent quality and we welcome further suggestions to improve them, both in terms of design and in terms of format. We are further working on redoing the figures in the appendix.
>
> Again we thank our reviewer for highlighting the value of the contribution, and for pointing out the accessibility issue with the figures.

---

> > ### Comment · Reviewer_L9hd · 2022-11-19
> > **Thanks for the rebuttal**
> >
> > Thanks for making the figures more accessible, however, I think they still are slightly hard to read with thin lines and small font sizes, I am sure you can make it look much nicer -- given the simple premise of the paper -- which needs to be conveyed across the community with ease.
> >
> > I will advocate for acceptance of the paper, but I still am expecting better illustrations and plots in this case!!

---

> > > ### Author Response · Authors · 2022-11-19
> > > **Agreed**
> > >
> > > Thank you for the careful check! We will revise the figures to ensure the font size is large enough and the lines are thick enough to ensure readability.

---

> > > ### Author Response · Authors · 2022-11-20
> > > **Updated Figures**
> > >
> > > [Here](https://pdfhost.io/v/NtHCYh78V_MindThePool) is an updated version of the manuscript where we improve both the accessibility and legibility of the figures.

---

### Official Review · Reviewer_JXFi · 2022-10-28

**Confidence:** 5
**Correctness:** 4
**Technical Novelty And Significance:** 2
**Empirical Novelty And Significance:** 3
**Recommendation:** 6

**Clarity, Quality, Novelty And Reproducibility:**

As mentioned in the previous section, the paper is clearly written and the quality of the figures certainly help in illustrating the analysis and presenting the results. Some minor elements that may be improved are the following:

* In the second paragraph of Section 3.1, note that "specially" is written twice instead of "spatially".
* The third paragraph of Section 3.2 starts with "It is evident that [...]" without referencing any figure or table.

The quality of the paper is good, as discussed before, and I am not aware of any code provided alongside the submission for reproducibility.

I have discussed novelty at length in the previous section.

**Strength And Weaknesses:**

### Strengths

The problem tackled by this paper is conceptually simple and therefore easy to follow. The paper is well written, the figures are very informative and the experimental design helps sheds light on the aspects that are analysed. Furthermore, the conclusions are supported by the results. Finally, the paper not only analyses a pitfall in the way convolutional neural networks are typically trained, but it also presents a method to overcome the problem. Therefore, I have an overall good impression of the paper.

### Weaknesses

My main concern is related to the novelty of the results presented in the paper. The paper seems to be an incremental extension of Alsallakh et al. (2021), which analysed spatial biases in CNNs due to padding. This paper sets the focus on the input size and the pooling operation ("mind the pool" vs. "mind the pad", in the titles), but fundamental issue underlying the pitfalls analysed in both papers is, my opinion, the same: padding. The issue with pooling is really padding, not pooling itself, as indeed explained by the authors. In fact, the problem seems to be identical regardless of the type of downscaling (due to pooling or striding), precisely because it is caused by padding. All elements---padding, input size, downscaling---are related, and in fact Alsallakh et al. (2021) already the effect of input size and pointed out at sharp differences when the input size differed by just one pixel.

Several elements of analysis in the present paper were already discussed in Alsallakh et al. (2021), for instance the asymmetry of the kernels (Figure 4 in "mind the pool" and Figure 6 in "mind the pad") or the shift of the receptive field (Figure 8 and 7 respectively). Furthermore, other previous work, such as Shocher et al. (2020) had set the focus specifically in the downscaling operation, as pointed out here. In this regard, a significant part of the analysis offers little novelty.

Unrelated to the above, one question I have regarding the proposed method to remove the bias towards the training image size is what the influence is of fixing one side for the padding at test time. I expect little influence, but having to stick to just one seems arbitrary and it would be nice to show experimentally whether the choice has no, little or moderate influence in the performance. As a suggestion, the curves in Figure 1 and equivalent figures could display a confidence interval obtained from testing with multiple options for padding at test time.

Finally, as a minor comment, when mentioning fully-convolutional architectures, I would cite Springenberg et al. (2015) (cited later in the paper) as well as Long et al. (2015). (The reviewer is not related to any of these papers).

**Summary Of The Paper:**

This paper provides an analysis of the effect of training convolutional neural networks with a particular image size in the generalisation to different image sizes at inference time, as a consequence of downscaling operations within the architecture and specifically the padding necessary for enabling flexible input sizes. The analysis demonstrates that standard methods exhibit drops in performance in a periodic fashion with the input size, as best illustrated in Figure 1, and provides an explanation for the effect. Further, the paper presents simple mechanism to overcome this issue: randomising the location of the padding during training to break the bias due to a systematic location, as is common practice.

**Summary Of The Review:**

My overall impression of this paper is positive, as it provides a comprehensive analysis of a pitfall in CNNs trained in the currently standard way, even if typically the input size is fixed as well during test time, as well as a simple enough solution to overcome the problem. Furthermore, the paper is well written. Nonetheless, it is also fair to point out that the novelty is limited in view of previous work such as Alsallakh et al. (2021) or Shocher et al. (2020), for instance.

---

> ### Author Response · Authors · 2022-11-16
> **Emphasizing the novelty, and inference with fixed padding**
>
> We thank our reviewer for the thorough review and for the helpful suggestions to improve our work.
> We agree that “all elements---padding, input size, downscaling---are related”. The key differences between our work (hereafter MPL) and the "Mind the Pad" paper (hereafter MPD) by Alsallakh et al. (2021) are:
> - While padding (and the boundary condition in general) is indeed a fundamental issue in both papers, MPD focuses mainly on analyzing how it induces blind spots and how to mitigate those, while MPL focuses mainly on generalizing to a wide range of input sizes.
> - MPD does demonstrate that training ImageNet classifiers on 225x225 images outperforms 224x224 images (+0.34 top-1 accuracy), and attributed the improvement to mitigating the skewness in the weights. While the authors aimed to control for the increased input size by introducing no new visual information, they did not control for the number of floating point operations: The larger input results in larger feature maps, and around 1% more operations during training. This renders the accuracy comparison rather unfair (interestingly, a +0.34 increase in accuracy is equivalent to a relative reduction of the top-1 error by ~1%). We did hint to this issue in footnote 2 on page 6.
> - In contrast, MPL introduces an adjustment to pooling layers (called SBPool) to mitigate potential skewness in the weights **without changing the input size**. Hence, all comparisons made are between models that use the same number of floating point operations. Furthermore, maintaining the input size is crucial, as it is not always feasible to change the input size: Resizing the input images incurs interpolation artifacts, padding them incurs padding artifacts, and cropping them incurs information loss.
> - SBPool generalizes significantly better to arbitrary input sizes compared with the alternatives explored in MPD including training on 225x225 inputs (Fig.1 vs Fig. 3-middle), other padding strategies (see Fig. 7), antialiasing (see Fig. 9-right), and padding-free pooling as in VGGNets (Fig. 18-right in Appendix D.3).
>
> It is also true that MPL reuses two visual analysis methods introduced in MPD, with proper citation. Our goal is to demonstrate how SBPool mitigates the skewness in the weights (Figure 4) and the misalignment in the receptive field (Figure 8) without changing the input size to 225x225 as in MPD. Finally, the common insight we have with Shocher et al. (2020) is that standard CNN can suffer from such misalignment (Figure 6 in their paper). We argued that CNNs can use this misalignment as a  cue about the input size, besides padding. Shocher et al Continuous Convolution layers mitigate the misalignment by learning subsampling operators that permit arbitrary non-integer input-to-output ratios, which differs substantially from SBPool.
>
> We thank our reviewer for suggesting to study the effect of fixing the padding side. We computed the accuracy after fixing the padding at all layers to the left and top sides at inference time. This resembles the behavior of standard CNNs. As expected, the accuracy generally drops, compared with the alternating scheme we propose (69.33 vs 69.64 for input size 224x224). Nevertheless, as a function of input size, the model with fixed padding sides does not exhibit sharp periodic drops in accuracy, because it was trained under SBPool. To further evaluate the effect, we recomputed the curve under 2^5 = 32 configurations, restricting the possibilities at each of the five downsampling layers to (u_h, u_w) = (0, 0) and (u_h, u_w) = (1, 1). [This](https://i.ibb.co/dfRbMMx/Figure-1.png) plot depicts the range of accuracy values at each input size. As a result, we find that fixing the padding to one side at inference time does have a mild impact on accuracy.
>
> We updated the introduction to cite both Long et al. (2015) and Springenberg et al. (2015) for fully-convolutional networks. We also fixed the other issues our reviewer thankfully pointed us to. We can further include some of the above-mentioned arguments to better highlight the contribution w.r.t. MPD and Shocher et al. (2020), and we can elaborate on the effect of fixing the padding at inference time in a new appendix if recommended by the reviewer.
>
> Again, we thank our reviewer for the helpful remarks, we hope we could address the issues raised and appreciate guidance on further changes needed to improve our manuscript.

---

### Comment · Area_Chair_7PDZ · 2022-11-18
**Responses**

Dear Reviewers,

Do you have any comments/replies to author's responses - it would be great if you could respond to them. Have they changed your opinion on the paper?

Kind regards,
AC

---

### Decision · Program_Chairs · 2023-01-20

**Decision:**

Accept: poster

**Justification For Why Not Higher Score:**

The method is nice and useful but not novel enough or of high enough importance to warrant higher score.

**Justification For Why Not Lower Score:**

Well done, with a simple method that works.

**Metareview: Summary, Strengths And Weaknesses:**

The paper considers a problem of applying a trained convolutional networks to images of different sizes of input than the one they have been trained on. They note the degradation in performance and carefully analyse and provide explanation for this problem as well as solution. The method is simple and easy to understand. Some writing can be improved as pointed out by the reviewers.

**Note From Pc:**

if the above contains the word "oral" or "spotlight" please see: "oral" presentation means -> notable-top-5% and "spotlight" means -> notable-top-25%. As stated in our emails, we are disassociating presentation type from AC recommendations